# Punicalagin Prevents Inflammation in LPS- Induced RAW264.7 Macrophages by Inhibiting FoxO3a/Autophagy Signaling Pathway

**DOI:** 10.3390/nu11112794

**Published:** 2019-11-15

**Authors:** Yuan Cao, Jihua Chen, Guofeng Ren, Yahui Zhang, Xiuying Tan, Lina Yang

**Affiliations:** Xiangya School of Public Health, Central South University, Changsha 410128, China; caoyuan2017@csu.edu.cn (Y.C.); chenjh@csu.edu.cn (J.C.); renguofeng@csu.edu.cn (G.R.); yahuizhang@csu.edu.cn (Y.Z.); iceytan@163.com (X.T.)

**Keywords:** punicalagin, inflammatory responses, FoxO3a/autophagy signaling pathway

## Abstract

Punicalagin, a hydrolysable tannin of pomegranate juice, exhibits multiple biological effects, including inhibiting production of pro-inflammatory cytokines in macrophages. Autophagy, an intracellular self-digestion process, has been recently shown to regulate inflammatory responses. In this study, we investigated the anti-inflammatory potential of punicalagin in lipopolysaccharide (LPS) induced RAW264.7 macrophages and uncovered the underlying mechanisms. Punicalagin significantly attenuated, in a concentration-dependent manner, LPS-induced release of NO and decreased pro-inflammatory cytokines TNF-α and IL-6 release at the highest concentration. We found that punicalagin inhibited NF-κB and MAPK activation in LPS-induced RAW264.7 macrophages. Western blot analysis revealed that punicalagin pre-treatment enhanced LC3II, p62 expression, and decreased Beclin1 expression in LPS-induced macrophages. MDC assays were used to determine the autophagic process and the results worked in concert with Western blot analysis. In addition, our observations indicated that LPS-induced releases of NO, TNF-α, and IL-6 were attenuated by treatment with autophagy inhibitor chloroquine, suggesting that autophagy inhibition participated in anti-inflammatory effect. We also found that punicalagin downregulated FoxO3a expression, resulting in autophagy inhibition. Overall these results suggested that punicalagin played an important role in the attenuation of LPS-induced inflammatory responses in RAW264.7 macrophages and that the mechanisms involved downregulation of the FoxO3a/autophagy signaling pathway.

## 1. Introduction

With increasing attention on healthy diets, people are increasingly oriented to seek a better lifestyle that is comprised of correct behaviors in dietary choices. Dietary guidelines around the world are based on the daily consumption of plant foods rich in phytochemicals that are able to cure diseases and attain a general state of health [1]. Epidemiological and scientific studies have reported that a high intake of fruits and vegetables rich in polyphenolic phytochemicals protect against diabetes mellitus, cerebrovascular diseases, neurodegenerative diseases, and arthritis [2,3,4].

Since ancient times, pomegranate (Punica granatum L.) has served as a “healing fruit” with numerous beneficial effects on several disease. In addition, pomegranates are usually utilized fresh and in processed forms such as juice, jam, and wine [1]. Pomegranate shows brilliant antioxidant [2], anticancer [3] and anti-inflammatory [4,5] properties attributed to its high ingredients of polyphenols including ellagitannins, ellagic acid, and other polyphenols [6,7]. Previous research has reported that punicalagin (PU), an ellagitannin that is the most abundant of these polyphenols, exhibits potent anti-inflammatory bioactivities, including inhibiting tumor necrosis factor (TNF) α, interleukin (IL) 1β or interleukin (IL) 6 production in lipopolysaccharide (LPS) stimulated RAW264.7 macrophages and primary human chondrocytes [8,9]. However, the precise molecular mechanisms explaining how PU suppresses the inflammatory response in macrophages are still not well understood.

Because inflammation is an important etiological factor common to a range of chronic diseases, this potential anti-inflammatory activity may be of critical importance to explain the apparent health-promoting activities of punicalagin, and therefore may also have great significance in the development of anti-inflammatory therapeutic agents. Macrophages are important inflammatory cells implicated in the initiation of inflammatory responses, which play critical roles in the pathogenesis of numerous inflammatory disease processes by secreting a great number of pro-inflammatory mediators and pro-inflammatory cytokines [10]. Lipopolysaccharide (LPS) plays an important role in inducing the inflammatory response and leading to various inflammatory diseases, and is a compound of the cellular wall of Gram-negative bacteria [11]. The interaction between LPS and toll-like receptor (TLR) 4 results in the activation of intracellular signaling through myeloid differentiation factor (MyD) 88 pathways, leading to the activation of major translocation of nuclear transcription factor κB (NF-κB) and its upstream regulators, mitogen-activated protein kinases (MAPKs), including p38, ERK1/2, and JNK [12,13]. Upon exposure to inflammatory stimulants, such as LPS, the free NF-κB translocated from cytoplasm into the nucleus, where the phosphorylated subunit p65 plays an important role by triggering transcription of certain genes, including TNF-α, IL-1β, and IL-6 [14]. Subsequently, these cytokines will further promote TLR4-mediated pathways activation, orchestrating the inflammatory response into a vicious circle [11].

Accumulating evidence has reported that autophagy is an important component of the innate immune response. Autophagy is an adaptive response of cells to metabolic stress and environmental changes, which degrades or removes damaged proteins and organelles in the cytoplasm through a lysosomal degradation pathway to maintain cell homeostasis [15]. Autophagy core protein LC3 exists as LC3I under normal conditions, and conversion from LC3I to LC3II can be observed when autophagy is activated [16]. The number of LC3II is clearly correlated with the amount of autophagosomes. In addition, the number of LC3II at a certain point does not determine autophagic flux, which is widely used to indicate overall autophagic degradation rather than autophagosome formation. Determination the expression of LC3II and p62. P62 is an autophagy adaptor and an important protein responsible for protein degradation, among other proteins, which has been used to detect the intracellular autophagic flux.

Therefore, it is critical to explore the amount of LC3II delivered to lysosomes by comparing LC3II and p62 levels with or without the lysosomal protease inhibitors, including chloroquine or bafilomycin A1 [17]. Beclin1 is a key molecule regulator that is required for vesicle nucleation during formation of autophagosome [18]. Autophagy has a wide range of immune functions, including cell autonomous defense and coordination of complex multicellular immune responses [19]. It was the first time to discover that autophagy could prevent excessive inflammatory reactions in ATG16L1-deficient mice, which are found to produce more inflammatory cytokines than normal wild-type mice and are more susceptible to sepsis [20]. Autophagy is a double-edged sword. For instance, with the application of autophagy inhibitors, the expression of inflammatory genes is enhanced in adipocytes; conversely, activation of autophagy reduces the expression of inflammatory genes, indicating that autophagy can balance the inflammatory response [21]. In contrast, with the autophagy inhibitors, 3-methyladenine blocked autophagy activity and reversed LPS-induced acute lung injury through the inhibition of inflammation and autophagy [22]. These observations indicated that the protective or detrimental effect of autophagy activation depends on cell types, stimuli, and environment, and the directionality of the effect may be context dependent. Therefore, it is necessary to figure out how autophagy works in LPS-induced RAW 264.7 macrophages, as well as what role autophagy plays in the anti-inflammatory effects of PU.

As reported above, PU has potent capacity to be used to treat many chronic inflammatory diseases, whereas the underlying mechanism reported is only the tip of an iceberg. A previous study reported that PU played a protective role in various kinds of diseases in vitro, including enhanced autophagic turnover that prevented cultured primary human syncytiotrophoblasts from apoptosis [23] and induced human U87MG glioma cell death through inducing both apoptotic and autophagic pathways [24]. Therefore, we speculate that autophagy may be responsible for the anti-inflammatory activity of PU. In this study, we try to explore the anti-inflammatory mechanism of PU in LPS-induced RAW264.7 macrophages, focusing on whether autophagy takes part in the inhibition of macrophage inflammatory response along with its possible mechanism.

## 2. Materials and Methods

### 2.1. Reagents and Antibodies

Antibodies against the following proteins were used: p-p65 (sc136548, Santa cruz Biotechnology, Inc. USA) P-JNK (sc293136, Santa cruz Biotechnology, Inc. USA); P-ERK1/2 (sc81492, Santa cruz Biotechnology, Inc. USA); p-p38 (sc7973, Santa cruz Biotechnology, Inc. USA); Beclin1 (11306-1-AP, Proteintech, Wuhan, China); LC3A/B (AF5402, Affinity Biosciences, USA); p62 (55274-1-AP, Proteintech, Wuhan, China); FoxO3a (10849-1-AP, Proteintech, Wuhan, China); β-actin (60008-1-Ig, Proteintech, Wuhan, China); horseradish peroxidase (HRP) conjugated goat anti-mouse (ZDR-5109, ZSGB-BIO Biotechnology Co. Ltd. Beijing, China), and horseradish peroxidase (HRP) conjugated goat anti-rabbit (ZB-5301, ZSGB-BIO Biotechnology Co. Ltd. Beijing, China). The followings reagents were used: punicalagin (65995-63-3; Chengdu Herbpurify Co. Ltd.Chengdu, China); lipopolysaccharide (LPS) from Escherichia coli 0111:B4 (L4391, Sigma-Aldrich, USA); chloroquine (C6628, Sigma-Aldrich, USA); and monodansylcadaverine (30432, Sigma-Aldrich, USA).

### 2.2. Cell Culture

RAW264.7, a mouse macrophage cell line (Xiangya central laboratory cell bank, Central South University, Changsha, China), was cultured in high-glucose Dulbecco’s modified Eagle’s medium (DMEM) (SH30022.01, HyClone, USA) containing 10% fetal bovine serum (FBS) (10099141C, Gibco, USA), 100 U/mL penicillin, and 100 μg/mL streptomycin (SV30010, HyClone, USA) at 37 °C in a humidified incubator containing 5% CO_2_.

### 2.3. Cell Viability Assay

After overnight culture in a 96-well plate (2 × 10^4^ cells/well, 100 μL medium/well), the cells were treated with punicalagin and lipopolysaccharide (LPS) for 24 h and 48 h, respectively. Thereafter, 20 μL Thiazolyl Blue Tetrazolium Bromide (MTT) (M8180, Solarbio, Beijing, China) solution was added to each well and incubated at 37 °C for 4 h. Finally, the absorbance of each well was recorded at 490 nm using a microplate reader (BioTek XS2, USA).

### 2.4. Nitric Oxide (NO) Production Assay

After overnight culture in a 24-well plate (1 × 10^5^ cells/well, 500 μL medium/well), the cells were pre-treated with punicalagin for 1 h and lipopolysaccharide for an additional 24 h, the culture supernatant from each well was collected at the end of scheduled experiments and used to measure NO production. On the basis of the Griess reaction, the NO production was determined using a commercial NO assay kit (S0021, Beyotime Institute of Biotechnology, Shanghai, China) according to the manufacturer’s instructions; 50 µL cell culture medium and 100 µL Griess reagents I and II were reacted in a 96-well plate at room temperature (RT) for 10 minutes, and the absorbance was then measured at 540 nm using a microplate reader.

### 2.5. ELISA of TNF-α and IL-6

After overnight culture in a 24-well plate (1 × 10^5^ cells/well, 500 μL medium/well), the cells were pre-treated with punicalagin for 1 h and lipopolysaccharide for an additional 24 h, the culture supernatant from each well was collected at the end of scheduled experiments and used to measure TNF-α (EK0527, BOSTER, Wuhan, China) and IL-6 (EK0411, BOSTER, Wuhan, China) concentration by ELISA according to the manufacturer’s instructions, and the absorbance was then measured at 520 nm using a microplate reader.

### 2.6. Monodansylcadaverine (MDC) Staining

After overnight culture in a 6-well plate (2.5 × 10^5^ cells/well, 1 mL medium/well), the cells were pre-treated with punicalagin for 1 h and lipopolysaccharide for an additional 24 h, after treatment, the cells were incubated with MDC (50 µmol/L) at 37 °C for 30 minutes. After incubation, the cells were washed two times with precooling PBS, and immediately observed and pictured under a fluorescence microscope (Invitrogen EVOS M7000, Germany).

### 2.7. Western Blotting

After overnight culture in a 6 cm plate (6 × 10^5^ cells/well, 3 mL medium/plate), the cells were pre-treated with punicalagin for 1 h and lipopolysaccharide for an additional 24 h, cells were harvested and lysed in lysis buffer containing 250 mM sucrose, 50 mM NaCl, 20 mM Tris-HCl, pH 7.4, 1 mM EDTA, 1% Triton X-100, 1 mM dithiothreitol, and 1 mM phenylmethylsulphonyl fluoride for 15 minutes on ice. After incubation, lysates were centrifuged (12,000 rpm, 10 minutes) and each supernatant was collected. BCA protein assay reagents (AR0146, BOSTER, Wuhan, China) were used to measure protein concentration. Samples were separated by SDS gel electrophoresis and electrotransferred onto polyvinylidene difluoride membrane in a wet electrophoretic transfer cell. Membranes were blocked with 5% nonfat dry milk powder dissolved in tris-buffered saline (20 mM Tris base, 0.5 M NaCl, pH 7.5) with 0.1% Tween-20 (TBST) for 1 h, at RT. The membranes were then incubated with primary antibodies overnight at 4 °C in a shaking incubator, washed with TBST 3 times and incubated for 1 h at room temperature with peroxidase-conjugated secondary antibodies diluted in blocking solution. After washing, proteins of interest were detected using BosterECL Star Western blotting detection reagent (AR1170, BOSTER, Wuhan, China).

### 2.8. Statistical Analysis

At least 3 independent experiments were performed for each condition, and the data were presented as means ± standard deviation (means ± SD) values. Statistical analysis was performed by either unpaired Student *t*-test or one-way analysis of variance analysis and statistical significance was evaluated using SPSS 18.0.

## 3. Results

### 3.1. Effects of PU on the Cell Viability of RAW264.7 Macrophages

To evaluate the cytotoxic effects of PU on RAW264.7 cells, cells were incubated with various concentration of LPS (0.001, 0.01, 0.1, 1, and 10 µg/mL) and PU (12.5, 25, 50, 100, 200, and 400 µM) for 24 h and 48 h. The MTT assay showed that LPS had no significant cytotoxic effects at concentrations up to 10 µg/mL at 24 h as compared with control cells that received no treatment (Figure 1A). Cell viability began to decrease to below 50% when the LPS concentration was increased to 10 µg/mL at 48 h. PU had no significant cytotoxic effects at concentrations up to 50 µM (Figure 1B). However, cell viability began to decrease to below 50% when the PU concentration was increased to 400 µM. Accordingly, we limited the concentration of PU in subsequent experiments to lower than 100 µM. In addition, we explored the release of LPS-induced NO, IL-6, and TNF-α on different concentrations (0.001, 0.01, 0.1, 1, and 10 µg/mL) and different time (6, 12, 24 h) (Appendix A) to find out the ideal acting condition. Finally, 1µg/mL LPS treatment for 24 h and 12.5, 25, 50 µM PU was chosen for further studies.

### 3.2. Effect of PU on Morphology of LPS-Induced RAW264.7 Macrophages

In this study, we first investigated the morphological changes of PU treatment in LPS-treated cells under optical microscopy. The cells were pre-treated with different concentrations of PU (12.5, 25, and 50 µM) for 1 h before adding LPS (1 µg/mL). The untreated control group RAW264.7 cells were round, with smooth cell edges without pseudopodia (Figure 2A), whereas those stimulated with LPS (1 µg/mL) for 24 h had characteristics of activation of macrophages, such as increase in cell size and elongated pseudopodia (Figure 2B). Following PU treatment, the changes in morphological structure of cells were ameliorated in a concentration-dependent manner (Figure 2C–E). PU treatment in the absence of LPS, the morphological changes are similar to the control group (Figure 2F). To evaluate the cytotoxic effects of PU on LPS-induced RAW264.7 cells, cells were pre-incubated with various concentrations of PU (12.5, 25, 50 µM) for 1 h and LPS (1 µg/mL) for an additional 24 h. The MTT assay showed that there were no significant cytotoxic effects under the treatment condition we used in our study (Figure 2G).

### 3.3. Effect of PU on NO Production and Pro-Inflammatory Cytokines Production in LPS-Induced RAW264.7 Macrophages

To investigate the anti-inflammatory of PU, LPS was used to stimulate the release of NO, IL-6, and TNF-α in the macrophage cells to mimic the chronic inflammatory environment. LPS exposure activated RAW264.7 cells inflammation reflection, as NO, IL-6, and TNF-α secretion in the supernatants significantly enhanced after LPS stimulation for 24 h, and pre-treatment with various concentrations of PU in prior to LPS challenge notably attenuated the enhancement of these cytokine secretions. The NO production was higher in the LPS group than in the control group. PU was found more potent to inhibit LPS-induced NO generation. (Figure 3A). LPS stimulation significantly upregulated the concentrations of pro-inflammatory cytokines IL-6, and TNF-α (Figure 3B,C). In contrast, treatment with PU at high concentrations (50 µM) significantly inhibited the levels of IL-6 and TNF-α that were induced by LPS. These results indicated that PU exerts anti-inflammatory activity via the suppression of NO production and pro-inflammatory cytokines IL-6 and TNF-α in LPS-induced RAW264.7 cells. The PU treatment alone had no effect on basal level of NO, IL-6, and TNF-α secretion in RAW 264.7 cells.

### 3.4. Inhibition of LPS-Induced NF-κB and MAPK Pathways Activation by PU

To further expound the mechanism of inhibition effect on LPS-induced pro-inflammatory cytokines secretion by PU, we then investigated the intervention of PU on LPS-stimulated activation of NF-κB and MAPKs signaling pathway. We assessed the effect of PU on LPS-stimulated phosphorylation of p65, JNK, ERK, and p38 using three different phospho-specific antibodies. Results indicated that as important signaling pathways in inflammation, p65, JNK, ERK, and p38 showed slight phosphorylation in cells of the control group. The LPS treatment significantly increased activation of NF-κB and MAPKs by strengthening phosphorylation of p65, JNK, ERK, and p38. The phosphorylation levels were attenuated to some degree in PU pre-treated cells as compared with LPS-stimulated cells (Figure 4A–D).

### 3.5. Effect of PU on Autophagy Inhibition in RAW 264.7 Macrophages

In addition, autophagy has been shown to be an important component of the innate immune response. To clarify whether autophagy was involved in the underlying mechanism of PU on LPS-stimulated RAW264.7 macrophages, we assessed the effect of PU on LPS-stimulated autophagy marker proteins LC3II, p62, and Beclin1 expression (Figure 5A–C). In comparison to the control, the LPS treatment decreased LCII and Beclin1 expression while it increased p62 expression significantly. In comparison to the LPS treatment, PU treatment enhanced LCII and p62 expression, decreased Beclin1 expression, which was consistent with LPS treatment. However, increased expression of the autophagy protein LC3II on its own does not necessarily indicate an increase in autophagic flux, since it also indicates an inhibition of autophagosome clearance. MDC, an autofluorescent agent, was proposed as a tracer for autophagolysosomes to analyze the autophagic process [25]. As shown in Figure 5D, the fluorescence intensity of LPS was lowest among these groups, and the fluorescence intensity of all groups was consistent with LC3II expression in Figure 5A–C. Similarly, the increase of autophagosomes was insufficient to reflex autophagosome maturation and degradation.

Thus, to determine if autophagy activity was inhibited or induced, cells were treated with autophagy inhibitor chloroquine (CQ), a specific late-phase autophagy inhibitor, which inhibits autophagic flux by preventing lysosomal degradation and blocks the fusion of autophagosomes with lysosomes, widely used to inhibit the maturation of autophagosomes into degradative autolysosomes [26]. We observed PU had an inhibition effect on LPS-induced IL-6 and TNF-α release at the highest concentration (50 µM), and the anti-inflammatory mechanism of PU was the key point, therefore, the concentration of PU at 50 µM was chosen for further experiments. As a rule, LC3II and p62 are widely used to monitor intracellular autophagy flux. We found that combination treatment with CQ as compared with the control group remarkably enhanced LC3II and p62 accumulation, whereas with LPS treatment, LC3II accumulation was less while p62 accumulation was more than the control group. LC3II and p62 expression in the PU and CQ combination group did not further aggravate as compared with PU supplementation alone (Figure 5E,F), these results indicating that LPS and PU both inhibited autophagy activation. The results worked in concert with LPS and the PU group decreased Beclin1 expression, one of the autophagy markers during autophagy initiation. In addition, the MDC-labeled autophagolysosomes indicated LC3II expression at some degree, as shown in Figure 5G, the degree of autophagolysosomes accumulated in the LPS group in the presence of CQ was less than the control group, and the fluorescence intensity of LPS combined with the PU group was unchanged in the absence or presence of CQ. These observations indicated that the fluorescence intensity showed the same tendency with LC3II expression. The morphological results further demonstrated that PU inhibits LPS-induced inflammatory responses dependent on autophagy inhibition.

To further prove the possibility, LPS-induced NO, IL-6 and TNF-α production in the presence of autophagy inhibitor CQ were explored and we observed that CQ suppressed LPS-induced NO, IL-6, and TNF-α release (Figure 6A–C), indicating that autophagy inhibition might be responsible for the suppressive effects of PU on LPS-induced secretion of pro-inflammatory cytokines in macrophage cells.

### 3.6. PU Suppresses Autophagy Via FoxO3a Signaling Pathway 

It is well known that FoxO3a is a potential regulator for autophagy [27,28]. To determine the signaling pathway responsible for the downregulation of autophagy genes following LPS treatment, a Western blot assay was performed to examine the levels of FoxO3a protein expression. We assessed the effect of PU on LPS-stimulated FoxO3a expression (Figure 7A). In comparison to the control, LPS and PU treatment both reduced FoxO3a expression, whereas PU treatment decreased FoxO3a expression stronger than LPS treatment. Meanwhile, the protein expression of FoxO3a and Beclin1 (as exhibited in Figure 5B) followed a similar trend. It was suggested that PU inhibited autophagy activation to suppress LPS-induced inflammatory response and was related to downregulate FoxO3a/autophagy signaling to some degree.

## 4. Discussion 

An increasing number of investigations have provided reliable evidence for the health-promoting properties of polyphenols. Polyphenols equipped with strong antioxidant, anti-inflammatory, and anticancer properties have been suggested to prevent the development and progression of chronic noncommunicable diseases, such as diabetes, Alzheimer’s disease, atherosclerosis diabetes, and cardiovascular diseases [29]. Polyphenol-rich food constituents, such as tea and wine, have been demonstrated to inhibit the development of inflammatory diseases and are also present in punicalagin. Here, we applied LPS to induce inflammation in RAW 264.7 macrophages to mimic a chronic inflammatory microenvironment in order to explore the anti-inflammatory activity of punicalagin, a natural polyphenolic compound from pomegranate, and the underlying mechanisms. Although we investigated the concentration in in vitro experiments, whether they were physiologically relevant was controversial. In most situations, it is difficult to precisely determine because in vivo metabolic studies are not available for various kinds of compounds. Punicalagin, an ellagitannin isolated from pomegranate polyphenols, is abundant in the fruit husk and juice in significant quantities, reaching levels of >2 g/L of juice, and accounts for more than one-fifth of the polyphenols in pomegranate juice [30]. Therefore, we hypothesized that the effects of punicalagin can reflect pomegranate juice to some degree and we found that the regulation of PU is linked to the inhibition of NF-κB, MAPK signaling activation, and FoxO3a/autophagy pathway regulation.

When challenged with LPS, macrophages are activated and release various pro-inflammatory factors and cytokines, and excessive release can result in extensive tissue damage and pathological changes [31,32]. Our results showed that PU at the highest concentrations used in our study strongly inhibited LPS-induced NO, TNF-α, and IL-6 secretion. In addition, MTT assay showed that the dosage of LPS and PU adopted in this study had no significant cytotoxic effects, indicating the anti-inflammatory effects of PU were not the result of poor cell viability. However, which targets could be affected by PU and played important roles in anti-inflammatory effects remained elusive. We found that PU attenuated LPS-induced phosphorylation of NF-κB, p38, JNK, and ERK MAPK, suggesting that PU suppressed NF-κB and MAPK signaling pathway to suppress the release of LPS-induced NO, TNF-α, and IL-6. Our current findings are in accordance with a previous study by Xu X et al. [8] that observed PU inhibited LPS-induced NF-κB and MAPK signaling activation.

Autophagy reduces pro-inflammatory signaling by eliminating damaged organelles in the cell, degrading pro-inflammatory signaling molecules, and controlling the production and release of inflammatory cytokines [33,34]. Autophagy can also eliminate or limit the communication of infection, as free microbes in the cytoplasm or microbes encapsulated by phagosomes can be captured and finally delivered by autophagosomes to lysosomal degradation [35,36]. Interestingly, LPS treatment exhibited a significant decrease in the levels of LC3II protein, whereas the LPS and PU combination treatment enhanced macrophages as compared with the untreated group (Figure 5A). Beclin1 proteins are required for autophagosome formation during autophagy [37]. Treatment with LPS alone, or in combination with PU, significantly suppressed the expression of autophagy marker protein Beclin1 (Figure 5C). Taken together, these results demonstrate that LPS treatment inhibits early stage autophagy and autophagosome accumulation, indicating that autophagy inhibition was insufficient to avoid the raised levels of inflammatory mediators. While under LPS and PU co-treatment condition, LC3II and Beclin1 expressions were increased and decreased, respectively, as compared with LPS treatment. These observations highlighted that the inhibition of autophagosome degradation brought about the contradiction between LC3II and Beclin1 expression. The probable reason to explain the phenomenon was that Beclin1 expression decreased, indicating autophagy was inhibited, and LC3II expression increased indicating autophagosome degradation was suppressed. P62 is an autophagy adaptor responsible for autolysosome degradation; autophagy inhibition results in a mass of accumulation of p62 followed by the formation of large aggregates for p62 and other ubiquitinated proteins. The LPS- and PU-induced p62 accumulation (Figure 5B) was in agreement with the assumption above, and the MDC assays also illustrated this point.

Therefore, we speculate that autophagy inhibition was strengthened in LPS-induced macrophages with pre-treatment of PU, along with the inhibition of inflammatory cytokines. To confirm this hypothesis, we used the autophagy inhibitor, CQ, to inhibit autophagy. Our results suggest that such autophagy inhibition led to a decrease in inflammatory cytokines, NO, TNF-α, and IL-6 secretion. Such data indicated that, under LPS-induced conditions, autophagy, by itself, was unable to prevent the inflammation. Interestingly, although PU and LPS both inhibited autophagy activation, PU enhanced the inhibition of autophagy on the basis of LPS, which gave rise to the opposite effect, i.e., pro-inflammation and anti-inflammation. Therefore, there is a dearth of research on the mechanism of autophagy activity in different environments or different stimuli. FoxO3a, which belongs to the FoxO family, is a transcription factor that regulates cell autophagy, survival, and senescence in mammals [38]. Accumulating evidence indicates that FoxO3a plays a crucial role by coordinating a program of autophagy activation [39]. Our data showed that FOXO3a downregulation inhibited autophagy to downregulate elevated inflammatory cytokines in LPS-induced RAW264.7 macrophages, which probably was a crucial target for the anti-inflammatory effects of PU.

Interestingly, accumulating evidence has reported that there is a complex network of MAPK, NF-κB signaling, and autophagy, and the mutual influences among them are still controversial and rely on cell type and stimuli. Mingxia Zhou et al. [40] found that LPS downregulates autophagy in colitis in vitro through the upstream TLR4-MyD88-MAPK signaling pathway, and, subsequently, orchestrates its downstream NF-κB activation, subsequently, resulting in the production of pro-inflammatory cytokines and oxidative stress. Feng-Ming Wang [41] reported that autophagy negatively regulated JNK phosphorylation activation and its downstream NF-κB signaling, and therefore autophagy inhibition played a crucial role in JNK-related inflammatory diseases. Alfredo Criollo et al. [42] reported that autophagic stimuli was required for the activation of NF-κB signaling pathway. Therefore, it is of great importance whether autophagy is involved in the anti-inflammation of PU in LPS-induced MAPK and NF- κB signaling activation and how autophagy works in this intersection.

## 5. Conclusions

In conclusion, our study provided new insight about the mechanisms of PU against LPS-induced inflammation. We identified that PU inhibited macrophage inflammation by inhibiting NF-κB, MAPK signaling pathway, and FoxO3a/autophagy signaling. Although further studies are warranted to elucidate the correlation between the regulation of autophagy and NF-κB and MAPK signaling pathways in macrophages, our findings showed that PU suppressed LPS-induced macrophage inflammatory responses potentially via the FoxO3a/autophagy signaling pathway. In summary, these results could be valuable for prevention and treatment of inflammatory diseases, as well as the development of PU and their use as a novel immune therapy in the treatment of inflammatory diseases.

On the basis of our results, future studies are required to further determine the underlying mechanism of punicalagin in different in vivo and in vitro models or even in the human body after consumption of PU from pomegranate juice, and therefore enrich the theoretical knowledge of PU and promote its future application.

## Figures and Tables

**Figure 1 nutrients-11-02794-f001:**
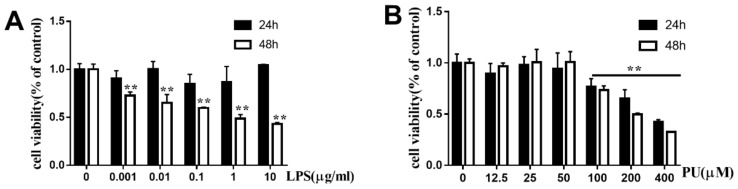
Cell viability of PU and LPS on RAW264.7 macrophages. (**A**–**B**) RAW264.7 cells were treated with various concentrations of PU and LPS for 24 h and 48 h. The cell viability was determined by MTT assay, as described in Materials and Methods. The data are presented as means ± SD (*n* = 3). (**, *p* < 0.01 vs. control group).

**Figure 2 nutrients-11-02794-f002:**
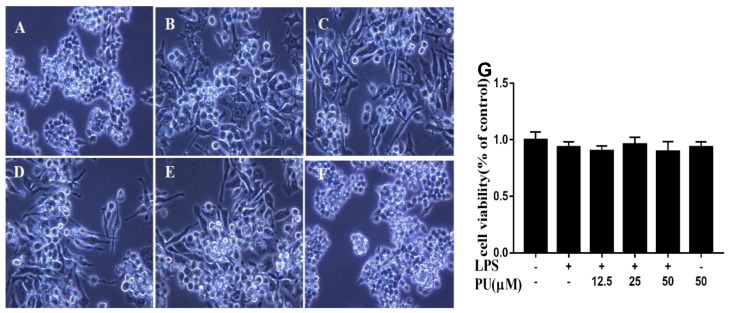
Photograph of RAW264.7 cells were pre-treated with various concentration of PU for 1 h and treated with LPS for an additional 24 h under optical microscopy (magnification x400). (**A**) Control; (**B**) LPS treatment; (**C**–**E**) LPS and PU (12.5, 25, and 50 µM) treatment; (**F**) PU treatment; and (**G**) cell viability of PU supplementation with LPS-induced RAW264.7 macrophages. The RAW264.7 cells were pre-treated with various concentrations of PU for 1 h and LPS for an additional 24 h. The cell viability was determined by MTT assay, as described in Materials and Methods. The data are presented as means ± SD (*n* = 3). (**, *p* < 0.01 vs. control group).

**Figure 3 nutrients-11-02794-f003:**
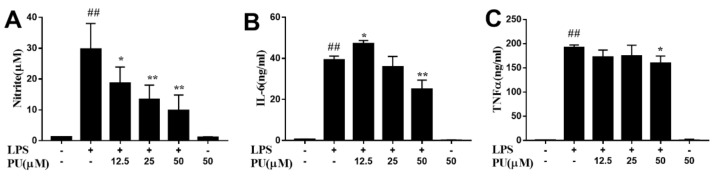
Anti-inflammatory effect of PU on LPS-induced RAW264.7 macrophages. (**A**–**C**) Cells were pre-treated with various concentrations of PU for 1 h and treated with LPS for an additional 24 h. The NO content was determined by Griess reagent and the production of cytokines were measured by enzyme-linked immunosorbent assay (ELISA) kit using the microplate reader. The data are presented as means ± SD (*n* = 3). (*, *p* < 0.05 and ** *p* < 0.01 vs. LPS group, and **##**, *p* < 0.001 vs. control group).

**Figure 4 nutrients-11-02794-f004:**
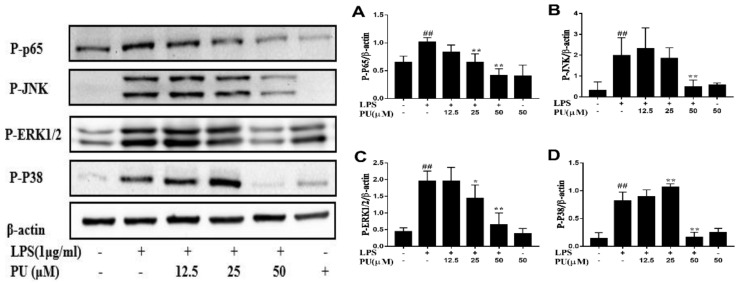
Inhibitive effects of PU on LPS-induced p65 (**A**), JNK (**B**), ERK1/2 (**C**) and p38 (**D**) phosphorylation in RAW264.7 macrophages. Cells were pre-treated with PU (12.5–50 μM) for 1 h before exposure to LPS (1 μg/mL) for 30 minutes. The data are presented as means ± SD (*n* = 3). (*, *p* < 0.05 and ** *p* < 0.01 vs. LPS group, and **##**, *p* < 0.01 vs. control group).

**Figure 5 nutrients-11-02794-f005:**
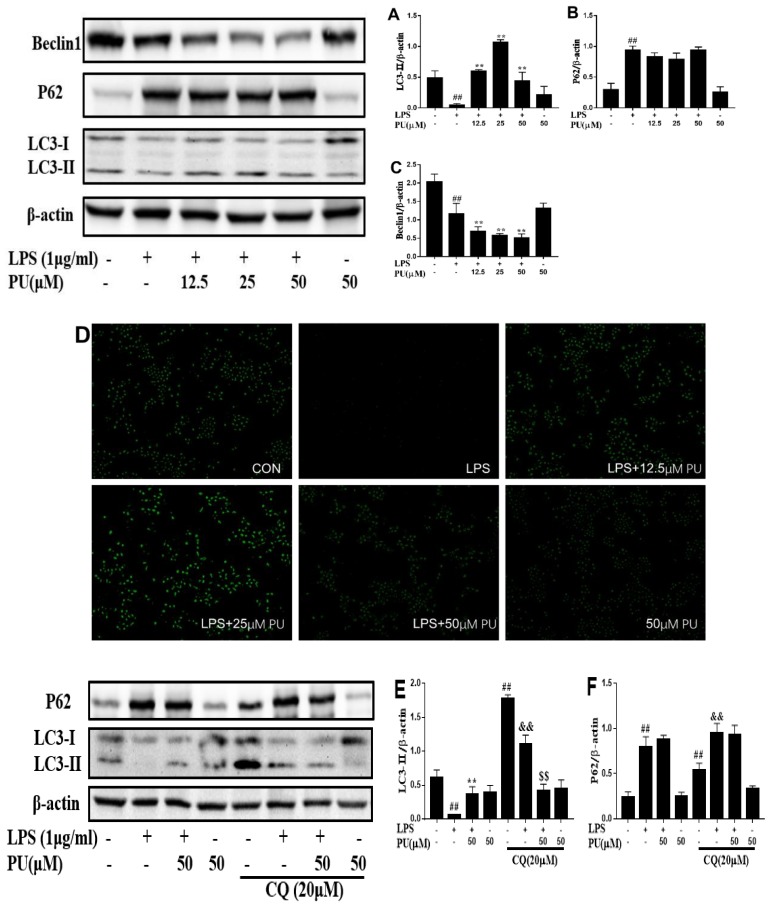
Effect of PU on autophagy inhibition in RAW 264.7 macrophages. (**A**–**C**) Cells were pre-treated with various concentrations of PU (12.5–50 μM) for 1 h and treated with LPS (1 μg/mL) for an additional 24 h. LC3II (**A**), p62 (**B**), Beclin1 (**C**) expression levels were determined by Western blot analysis. Autophagolysosomes were determined by MDC assays (**D**) under a fluorescence microscope (magnification x200). Representative image from three independent experiments has been shown along with β-actin as internal loading control. (**E** and **F**) Cells were pre-treated with chloroquine (20 μM) for 2 h, followed by treatment with PU (50 μM) for 1 h and treated with LPS (1 μg/mL) for an additional 24 h. LC3II (**E**), p62 (**F**) protein level were examined by Western blot analysis as described previously. Autophagolysosomes was determined by MDC assays (**G**) under a fluorescence microscope (magnification x200). Images are representative of three independent experiments that showed similar results. The data are presented as means ± SD (*n* = 3). (**, *p* < 0.01 vs. LPS group; ##, *p* < 0.01 vs. control group; &&, *p* < 0.01 vs. control with CQ treatment group; $$, *p* < 0.01 vs. LPS with LPS group).

**Figure 6 nutrients-11-02794-f006:**
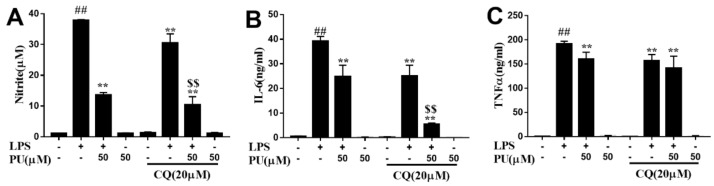
Anti-inflammatory effect of PU on LPS-induced RAW264.7 macrophages. (**A**–**C**) Cells were pre-treated with chloroquine (20 μM) for 2 h, followed by treatment with PU (50 μM) for 1 h and treated with LPS (1 μg/mL) for an additional 24 h. The NO content was determined by Griess reagent and the production of cytokines were measured by enzyme-linked immunosorbent assay (ELISA) kit using the microplate reader. The data are presented as means ± SD (*n* = 3). (*, *p* < 0.05 and **, *p* < 0.01 vs. LPS group; #, *p* < 0.05 and ##, *p* < 0.01 vs. control group; $, *p* < 0.05 and $$, *p* < 0.01 vs. LPS combined with CQ group).

**Figure 7 nutrients-11-02794-f007:**
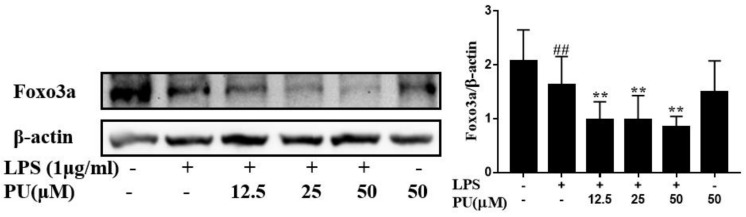
Effects of PU (**A**) on LPS-induced Foxo3a in RAW264.7 macrophages. Cells were pre-treated with PU (12.5–50 μM) for 1 h before exposure to LPS (1 μg/mL) for 30 minutes. Representative image from three independent experiments are shown along with β-actin as the internal loading control. The data are presented as means ± SD (*n* = 3). (**, *p* < 0.01 vs. LPS group; and **##**, *p* < 0.01 vs. control group).

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
