# Peer review of "Punicalagin Prevents Inflammation in LPS- Induced RAW264.7 Macrophages by Inhibiting FoxO3a/Autophagy Signaling Pathway"

_nutrients, 2019, doi:10.3390/nu11112794_

Round 1

Reviewer 1 Report

The manuscript describes putative beneficial effects of punicalagin, a polyphenol abundant in pomegranate juice, on cultured macrophages. The anti-inflammatory activity of PU had been previously reported. The current study provides data regarding the inhibition of autophagy promoted by PU and suggests a putative relationship between this observation and the diminution in inflammation.

Points to address:

-          Graphs cannot be clearly understood because the font size is very small 

-          It is important to discuss how the knowledge resulting from this in vitro study results in benefits of dietary phytochemical. PU anti-inflammatory effect is only observed at the highest PU concentration, and versus LPS at high concentration. According to this information, do the results support the benefits of pomegranate juice intake in healthy subjects?

-          The anti-inflammatory activity of PU via the suppression of MAPK and NFkB had been previously described (reference 8). The current study suggests that this action of PU is mediated by autophagy inhibition. In my opinion, it should be corroborated by additional experiments (i.e. microscopy studies to detect cell structures characteristics of autophagy,  or measurement of other proteins related to autophagy)

-          There is a close interplay among ER stress, apoptosis and autophagy. Did the authors measure some parameter in this regard?

-          The putative relationship between NF-kB, p38MAPK, and autophagy should be discussed with reference to previous bibliography

-          A limitation of the study is the low number of experiments.

-          The use of another house-keeping (different from b-actin) is recommended to avoid modifications in expression by cytoskeletal changes.

-          Methods should be clearly detailed. Please, indicate in each determination the number of cells, concentration of the stimuli, pre-incubation or co-incubation condition. Indicate the objective used in microscopy capture images.

-          I think there is a mistake when you state that viability decrease when PU concentration was 10 ug/ml (line 157).

Author Response

Dear professor,

    Many thanks for your precious comments on our paper, we have revised our paper according to your comments uploaded as an attached file. Thank you again for giving me so much precious advice to make our experiment design more comprehensive and more rigorous, and make our results more convincing. At the same time, thank you for pointing out some details and mistakes to us, and we will be more careful in future.

Kind regards,

Yuan Cao

E-mail: caoyuan2017@csu.edu.cn

Reviewer 2 Report

Article: Nutrients-613580

Title: Punicalagin prevents inflammation in LPS-induced RAW264.7 macrophages through inhibiting FoxO3a/autophagy signaling pathway

The manuscript gives an insight into the anti-inflammatory properties displayed by punicalagin in an in vitro model of inflammation.  The manuscript is readable and could be of interest for the readership of the journal. However major revisions are required.

Major:

1) The authors tested the effect of different concentrations of LPS and Punicalagin on the Cell Viability of RAW264.7 macrophages independently at two time points. However, they have no indication that the working conditions used further, i.e. 1μg/mL of LPS in combination with Punicalagin (up to 50 μM), are not cytotoxic.      

2) In Figure 5A, Punicalagin at the concentration of 25 μM (and in the presence of 1μg/mL of LPS) induced the highest expression LC3â…¡. For further experiments (in the presence of Chloroquine 20 μM, Figure 5C) Punicalagin at the concentration of 50 μM was used. Authors should show the effect of Punicalagin (25 μM) with Chloroquine 20 μM.

3) What is the effect of Punicalagin in the presence of Chloroquine on Beclin 1 expression?

Minor:

Please change “dose” with “concentration“ in the Abstract and throughout the manuscript.

Author Response

Dear professor,

Many thanks for your precious comments on our paper, we have revised our paper according to your comments uploaded as an attached file. Thank you again for giving me so much precious advice to make our experiment design more comprehensive and more rigorous, and make our results more convincing. All these benefits me a lot.

Kind regards,

Yuan Cao

E-mail: caoyuan2017@csu.edu.cn

Round 2

Reviewer 1 Report

The authors have addressed the major part of my questions. Regarding autophagy, the addition of results provides information and supports the conclusions. However, I still have some concerns.

I recommend revise English language and style, particularly to improve the new added paragraphs, because some parts of the text are difficult to follow.

The results about autophagy are difficult to understand. The text added in lines 273-310 is too long. As a suggestion, methodological details could be avoided to shorten the explanation. In my opinion, a graphical abstract could help to make clear the relationship between the pathways altered by PU addition and to better understand the results.

Regarding my previous question 2, I think that some comment about the equivalence of PU dose used in the study and the estimated concentration in pomegranate juice should be added in Discussion. Limitations of the study should also be included in Discussion

Please, define MDC the first time that it appears in the text.

Author Response

Dear professor,

Thaks for your precious recommendations, we have done some mild modifications for our manuscipt.Please see the attachment.

Kind regards,

Yuan Cao

E-mail: caoyuan2017@csu.edu.cn

Reviewer 2 Report

In the revised manuscript, the authors addressed all the major concerns.

As regards the minor comments, they changed "dose" into "concentration", as suggested. However, they also make the other way around, i.e. "concentration" into "dose". I would suggest to use "concentration" throught the manuscript, since "dose" is usually used in in vivo studies. 

Please also correct "resspectively" line 130, page 3

Author Response

Dear professor,

    Thanks for your precious recommendations, we have done some mild modifications for our manuscript. Please see the attachment.

Kind regards,

Yuan Cao

E-mail: caoyuan2017@csu.edu.cn